# EHMT2 epigenetically suppresses Wnt signaling and is a potential target in embryonal rhabdomyosarcoma

**Ananya Pal[1], Jia Yu Leung[1,2], Gareth Chin Khye Ang[1,2], Vinay Kumar Rao[1], Luca Pignata[2,3], Huey Jin Lim[4], Maxime Hebrard[2], Kenneth TE Chang[5], Victor KM Lee[4], Ernesto Guccione[2,3], Reshma Taneja[1]\***

[1]Department of Physiology, Yong Loo Lin School of Medicine, National University of Singapore, Singapore, Singapore; [2]Institute of Molecular and Cell Biology (IMCB), Agency for Science, Technology and Research (A*STAR), Singapore, Singapore; [3]Department of Biochemistry, Yong Loo Lin School of Medicine, National University of Singapore, Singapore, Singapore; [4]Department of Pathology, Yong Loo Lin School of Medicine, National University of Singapore, Singapore, Singapore; [5]Department of Pathology, KK Women and Children's Hospital, Singapore, Singapore

**Abstract** Wnt signaling is downregulated in embryonal rhabdomyosarcoma (ERMS) and contributes to the block of differentiation. Epigenetic mechanisms leading to its suppression are unknown and could pave the way toward novel therapeutic modalities. We demonstrate that EHMT2 suppresses canonical Wnt signaling by activating expression of the Wnt antagonist *DKK1*. Inhibition of EHMT2 expression or activity in human ERMS cell lines reduced *DKK1* expression and elevated canonical Wnt signaling resulting in myogenic differentiation in vitro and in mouse xenograft models in vivo. Mechanistically, EHMT2 impacted Sp1 and p300 enrichment at the *DKK1* promoter. The reduced tumor growth upon EHMT2 deficiency was reversed by recombinant DKK1 or LGK974, which also inhibits Wnt signaling. Consistently, among 13 drugs targeting chromatin modifiers, EHMT2 inhibitors were highly effective in reducing ERMS cell viability. Our study demonstrates that ERMS cells are vulnerable to EHMT2 inhibitors and suggest that targeting the EHMT2-DKK1-β-catenin node holds promise for differentiation therapy.

*For correspondence:
phsrt@nus.edu.sg

Competing interests: The authors declare that no competing interests exist.

## Introduction

Rhabdomyosarcoma (RMS) is the most common malignant soft tissue sarcoma (*Hawkins et al., 2013*; *Keller and Guttridge, 2013*; *Hettmer et al., 2014*) that arises due to a block in myogenic differentiation. Children with high risk disease have poor prognosis with only 30% showing 5-year event-free survival. Embryonal rhabdomyosarcoma (ERMS) accounts for the majority (~60%) of all RMS cases. No single genetic lesion is linked to ERMS but chromosome gains (chr 2, 8, 12, and 13) and loss of heterozygosity at 11p15.5 are characteristically seen (*Shern et al., 2014*). A few recurrent mutations occur in ERMS that include mutations in p53 (TP53), amplification of *CDK4*, upregulation of *MYCN*, and point mutations in RAS leading to its activation (*Hawkins et al., 2013*; *Shern et al., 2014*; *Zhu and Davie, 2015*; *Skapek et al., 2019*). Recent studies have investigated whether improper epigenetic imprinting underlies the myogenic differentiation defect in RMS (*Cieśla et al., 2014*). This includes altered expression of histone deacetylases, methyltransferases as well as lncRNAs and microRNAs that inhibit differentiation. Among these, EZH2 that mediates repressive histone H3 lysine 27 trimethylation (H3K27me3) is upregulated and binds to muscle specific genes in

ERMS. Its silencing increases both MyoD binding and transcription of target genes (*Ciarapica et al., 2014*). Similarly, HDAC inhibitors have been found to induce differentiation and reduce self-renewal and migratory capacity of ERMS by regulating Notch-1 and EphrinB1-mediated pathways (*Vleeshouwer-Neumann et al., 2015*). Interestingly, overexpression of the lysine methyltransferase SUV39H1 suppresses tumor formation in $KRAS^{G12D}$-driven zebrafish model of ERMS (*Albacker et al., 2013*).

Genetic mouse models that develop ERMS-like tumors due to deregulation of key signaling pathways such as Hedgehog, Wnt, Notch, and Yap signaling have been described (*Pal et al., 2019*). Double mutants lacking p53 and c-fos (Trp53$^{-/-}$/Fos$^{-/-}$) develop ERMS. Elevated expression of Wnt antagonists dickkopf-related protein 1 (*DKK1*) and secreted frizzled-related proteins (*sFRPs*), as well as downregulation of Wnt agonists such as Wnt ligands Wnt 7b, Wnt 5a, Wnt four, and Wnt 11 were reported in these tumors (*Singh et al., 2010*). Mice expressing activated smoothened under the *Fabp4* promoter leading to activation of Hedgehog signaling also develop ERMS. The tumors also show upregulation of *Dkk3* which, similar to *Dkk1*, inhibits canonical Wnt signaling (*Hatley et al., 2012*). Consistent with these findings, GSK3β inhibitors that activate Wnt signaling were most effective inducers of differentiation in a zebrafish model of ERMS (*Chen et al., 2014*). Importantly downregulation of Wnt signaling was found to be relevant only in ERMS.

There are 19 Wnt ligands, which function either through the canonical or non-canonical pathways in a highly context-dependent manner (*Masuda and Ishitani, 2017*). Canonical signaling is activated when Wnt ligands bind to a receptor from the Frizzled (Fzd) family. The co-receptors LRP5/6 facilitate Wnt signaling. Activation of Wnt signaling leads to disruption of the destruction complex (APC, Axin, GSK3β, and CK1α) which phosphorylates and degrades β-catenin. Induction of Wnt signaling results in accumulation of non-phosphorylated β-catenin (active β-catenin). β-catenin then translocates to the nucleus where it activates genes in cooperation with TCF/LEF1, but also other transcription factors (*Masuda and Ishitani, 2017*). Neither of the two non-canonical pathways (Planar Cell Polarity pathway [PCP] and the Wnt/calcium signaling pathway) involve β-catenin. Dkk1, a secreted protein, interacts with Lrp5/6 and antagonizes Wnt signaling by preventing Lrp5/6 association with Wnt/Fzd complex (*Niehrs, 2006*). Despite the relevance of Wnt signaling in ERMS, epigenetic mechanisms leading to its suppression have not been described and could pave the way to development of targeted therapies.

EHMT2, a lysine methyltransferase that is encoded by the *EHTM2* gene, mediates mono and di-methylation of H3K9 (H3K9me1/2), which is primarily involved in transcriptional repression (*Shinkai and Tachibana, 2011*). Recent studies however have shown that EHMT2 can also function as an activator in methylation-independent and -dependent ways (*Shankar et al., 2013*; *Casciello et al., 2015*). EHMT2 has been proposed to have oncogenic functions and its overexpression in leukemia, gastric, lung, prostate cancer, and alveolar RMS causes silencing of tumor suppressor genes through its H3K9me2 activity (*Shankar et al., 2013*; *Casciello et al., 2015*; *Bhat et al., 2019*). In this study, we found that canonical Wnt/β-catenin signaling is epigenetically suppressed in ERMS. EHMT2 activates expression of *DKK1* in a methylation-dependent manner through an impact on Sp1 and p300 recruitment. Our data indicate the potential of targeting the EHMT2-DKK1 axis to activate Wnt signaling for the development of novel ERMS therapeutics.

## Results

### EHMT2 inhibitors reduce ERMS cell viability

We recently reported that *EHMT2* is overexpressed in ARMS (*Bhat et al., 2019*). To examine whether EHMT2 expression is de-regulated in ERMS, and if it is functionally relevant in these tumor subtype, we first examined its expression in 16 ERMS patient tumor sections. High nuclear expression relative to normal muscle was apparent (*Figure 1A*). In addition, compared to primary human skeletal muscle myoblasts (HSMMs), EHMT2 overexpression at both mRNA and protein levels was apparent in three ERMS patient-derived cell lines RD18, JR1, and RD (*Figure 1B and C*). To examine if the EHMT2 pathway is functionally relevant, we treated JR1 and RD cell lines with 13 methyltransferase inhibitors at three different concentrations. Viability was measured 8 days later using MTS assay. Drugs targeting BRD4, PRMT5, and EHMT2 showed a strong effect on viability of both cell lines (*Figure 1D and E*). Strikingly, UNC0642 showed significantly higher efficacy against RD cells

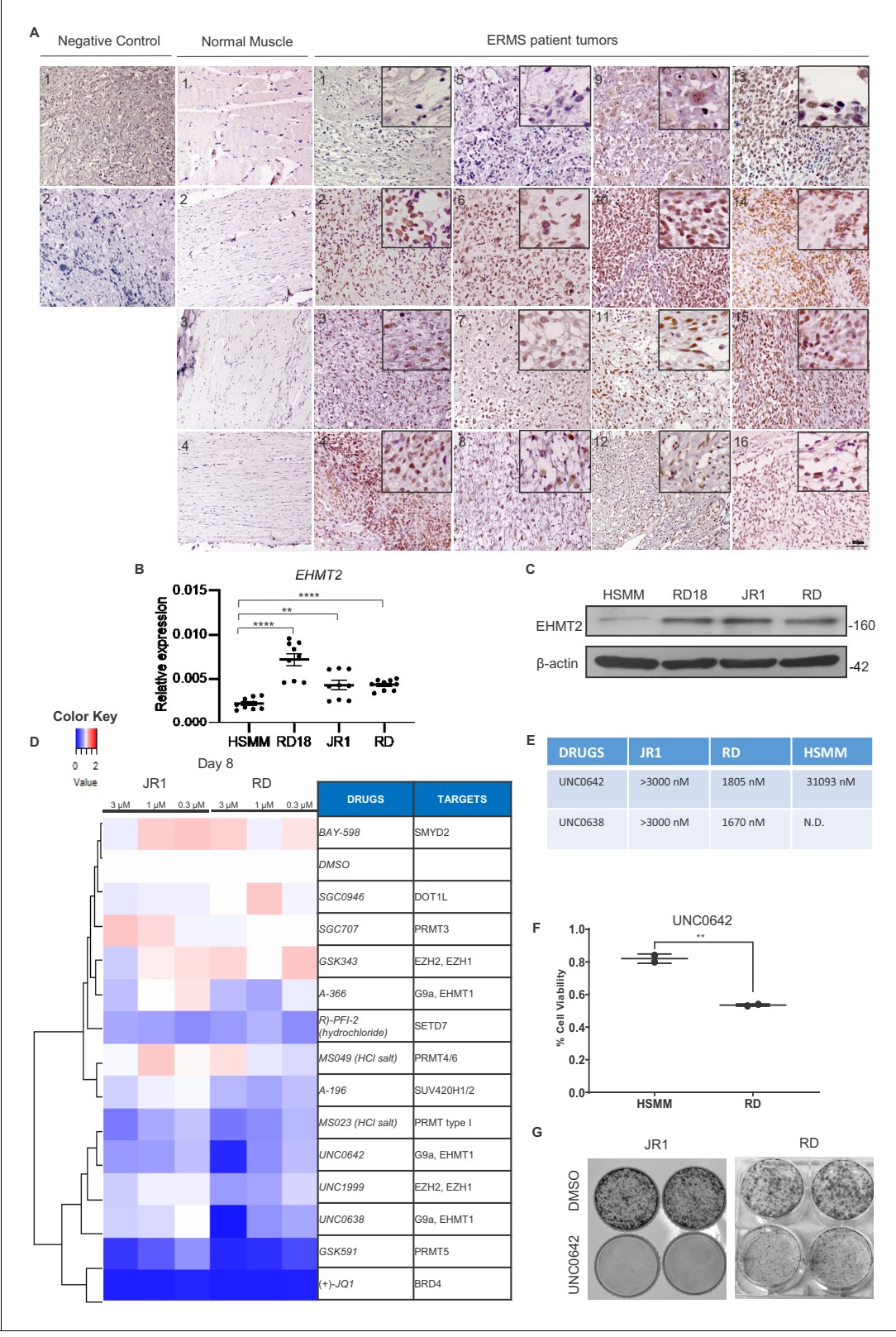

**Figure 1.** EHMT2 is overexpressed in embryonal rhabdomyosarcoma (ERMS). (**A**) 16 archival ERMS patient tumor specimens and four normal muscle samples were analyzed by immunohistochemistry using anti-EHMT2 antibody. Negative control indicates staining using secondary antibody alone. Inset shows zoomed in image of nuclear EHMT2 staining. Scale bar: 100 μm. (**B**) *EHMT2* mRNA (n = 3) were examined in three patient-derived cell lines (RD, RD18, and JR1) in comparison to primary human skeletal muscle myoblasts (HSMMs) by qPCR. Values correspond to the average ± SEM. All three

*Figure 1 continued on next page*

*Figure 1 continued*

ERMS cell lines examined showed an increased *EHMT2* mRNA expression compared to HSMM. (C) EHMT2 protein levels were examined by western blotting in HSMM, RD18, JR1, and RD cells. A representative image of three different experiments is shown. All three ERMS cell lines examined showed an increased EHMT2 protein expression compared to HSMM. (D) ERMS cell lines JR1 and RD were treated with the indicated methyltransferase inhibitors (3, 1, and 0.3 µM). Viability on day 8 was scored by MTS assay and measured as the ratio over control cells treated with an equivalent dilution of DMSO. RED indicates viability >control; WHITE is equal to control, and BLUE is less than control. The experiment was conducted in triplicates and (+)-JQ1 was used as a positive control. GSK591, UNC0642, and UNC638 had a strong effect on viability. (E) The IC50 of EHMT2 inhibitors in JR1, RD, and HSMM is shown. (F) HSMM and RD cells were treated with DMSO or UNC0642 for 6 days. Cell viability was assessed by trypan blue staining. (G) JR1 and RD cells were treated with DMSO or UNC0642 for 9 days. Colony formation was assessed by staining with crystal violet. A representative image of three different experiments is shown. In (B) data from three independent biological replicates each with three technical replicates were plotted. Statistical significance was calculated by unpaired two-tailed *t*-test. **p≤0.01, ***p≤0.001. N.D. = not determined.

The online version of this article includes the following source data for figure 1:

**Source data 1.** qPCR data for endogenous G9a expression in ERMS cell lines.

when compared to HSMM in cell viability assays (*Figure 1F*). Consistent with our drug screening assay, treatment of both JR1 and RD cells with UNC0642, a small molecule inhibitor of EHMT2 led to a striking reduction in colony formation (*Figure 1G*). Together, these results indicate that EHMT2 is overexpressed and functionally relevant in ERMS.

## EHMT2 inhibition promotes myogenic differentiation and inhibits proliferation in ERMS cell lines

To examine the role of EHMT2 in ERMS, we depleted its endogenous expression in RD18, JR1, and RD cells using small interfering RNA (*Figure 2A*, *Figure 2—figure supplement 1A*, and *Figure 2—figure supplement 2A*), or blocked its methyltransferase activity using UNC0642 that resulted in reduced H3K9me2 (*Figure 2B*, *Figure 2—figure supplement 1B*, and *Figure 2—figure supplement 2B*). We then examined the impact on differentiation and proliferation of tumor cells. EHMT2 knockdown (siEHMT2 cells) as well as UNC0642 treatment resulted in increased myogenic differentiation relative to their respective controls as evidenced from the increased MHC expression, a terminal differentiation marker, as well as *MYOG*, an early differentiation marker (*Figure 2C*, *Figure 2—figure supplement 1C*, and *Figure 2—figure supplement 2C*). To differentiate, myoblasts irreversibly exit the cell cycle (*Kitzmann and Fernandez, 2001*). Given the enhanced myogenic differentiation upon EHMT2 depletion, we investigated the impact of EHMT2 loss on proliferation by labeling S-phase cells with BrdU. Both siEHMT2 cells and UNC0642 treatment resulted in a significant decrease in BrdU$^+$ cells compared to their respective control in RD18 cells (*Figure 2D*, *Figure 2—figure supplement 1D*, and *Figure 2—figure supplement 2D*). Further, stable EHMT2 knockdown in RD cells also resulted in increased MHC levels and decreased BrdU$^+$ cells (*Figure 2—figure supplement 2E–G*). A striking reduction in colony formation was also seen in shEHMT2 RD cells compared to controls (*Figure 2—figure supplement 2H*). Together, these results indicate that EHMT2 inhibition permits cells to exit the cell cycle and undergo myogenic differentiation.

## EHMT2 regulates DKK1 and canonical Wnt signaling

In order to identify mechanisms underlying EHMT2 function, we performed RNA-Sequencing (RNA-Seq). Cluster analysis of differentially expressed genes from control RD and EHMT2 knockdown RD cells was done in triplicates (*Figure 3A*). Volcano plot of differentially expressed genes (*Figure 3B*) revealed that 872 genes were significantly upregulated in siEHMT2 cells compared to the control, of which 494 genes had a fold change >1.2. Among the 1098 genes that were significantly downregulated in siEHMT2 cells, 695 genes had a fold change >1.2. Gene ontology (GO) analysis showed that among the top 20 unique biological processes associated with differentially expressed genes in siEHMT2 cells were cell cycle progression and Wnt signaling (*Figure 3C*). Given its relevance in ERMS, we focused on the Wnt pathway. Interestingly, negative regulators of the Wnt pathway such as *DKK1*, *DKK3*, and *ITGA3* (*Shukrun et al., 2014*; *Kato et al., 2011*) were downregulated in siEHMT2 cells, whereas positive regulators such as *WNT3* and *FRAT2* were upregulated (*Figure 3—figure supplement 1A and B*). We validated genes in the Wnt pathway (*Figure 3D* and *Figure 3—figure supplement 1C–F*) as well as those involved in skeletal muscle differentiation such as

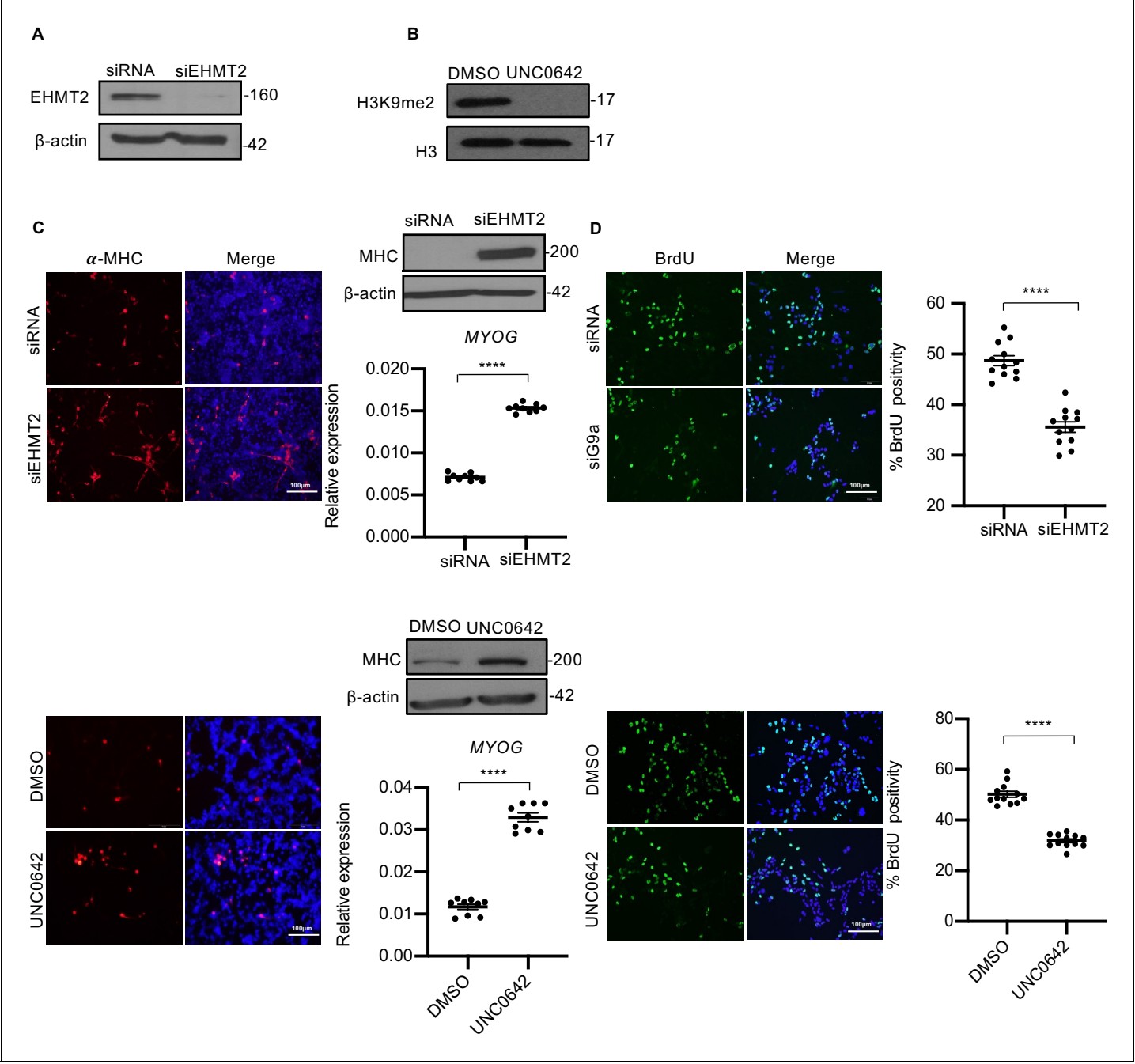

**Figure 2.** EHMT2 inhibits differentiation and promotes proliferation of myoblasts. (**A**) EHMT2 was depleted in RD18 cells using siRNA. Control and siEHMT2 cells were analyzed for knockdown efficiency by western blot. β-actin was used as an internal loading control. (**B**) H3K9me2 levels were analyzed 48 hr after 2.5 μM of UNC0642 treatment in RD18 cells. Histone H3 was used as a loading control. (**C**) Differentiation was analyzed in control and siEHMT2 RD18 cells (upper panels) or DMSO and 2.5 μM of UNC0642 RD18-treated cells (lower panels) after culture for 5 days in differentiation medium (DM). Cells were analyzed by immunofluorescence and western blot using anti-MHC antibody as indicated. Nuclei were stained with DAPI. Representative images of three different experiments are shown. Expression of *MYOG* was analyzed by qPCR at day 2 of differentiation (n = 3). Values correspond to the average ± SEM. (**D**) Proliferation was analyzed in control and siEHMT2 RD18 cells (upper panels); or DMSO and 2.5 μM of UNC0642-treated RD18 cells (lower panels) by immunostaining with anti-BrdU antibody. Cells were analyzed by immunofluorescence (n = 3). The dot plots show the percentage of BrdU[+] in siEHMT2 and UNC0642-treated cells relative to their respective controls. Values correspond to the average ± SEM. In (**C**) data from three independent biological replicates each with three technical replicates were plotted. In (**D**) data from three independent biological replicates each with four technical replicates were plotted. Statistical significance was calculated by unpaired two-tailed *t*-test. ****p≤0.
The online version of this article includes the following source data and figure supplement(s) for figure 2:

**Source data 1.** qPCR data for day 2 myogenin expression in RD18 cells upon G9a knockdown.

*Figure 2 continued on next page*

*Figure 2 continued*

**Source data 2.** qPCR data for day 2 myogenin expression in RD18 cells upon G9a activity inhibition by UNC0642.
**Source data 3.** BrdU quantification data in RD18 cells upon G9a knockdown.
**Source data 4.** BrdU quantification data in RD18 cells upon G9a activity inhibition by UNC0642.
**Figure supplement 1.** Loss of EHMT2 expression or activity in JR1 cells increases differentiation and reduces proliferation.
**Figure supplement 2.** Loss of EHMT2 expression or activity in RD cells increases differentiation and reduces proliferation.

*MYOD1*, *MYOSTATIN*, *MYL*, and *MYOZENIN* by qPCR in RD18 cells (*Figure 3—figure supplement 1G–J*).

DKK1 is a member of the Dickkopf family that inhibits canonical Wnt/ß-catenin signaling by binding to and inhibiting the Wnt co-receptor LRP5/6 (16). Consistent with the transcriptomic data, downregulation of *DKK1* mRNA was apparent in RD18 siEHMT2 cells compared to control cells by qPCR (*Figure 3D*). Interestingly UNC0642 treatment also resulted in a decrease in *DKK1* mRNA expression in RD18 cells (*Figure 3E*). DKK1 protein levels also decreased in both siEHMT2 and UNC0642 RD18-treated cells (*Figure 3F and G*). Moreover, the reduction in DKK1 levels correlated with increased active-ß-catenin in siEHMT2 cells (*Figure 3H*) in RD18, RD shEHMT2 cells (*Figure 3I*) and upon UNC0642 treatment in RD18 (*Figure 3J*) when compared to their respective controls. These results indicate that loss of EHMT2 leads to downregulation of DKK1 with concomitant activation of canonical Wnt signaling and myogenic differentiation.

## EHMT2 regulates DKK1 through Sp1/p300 occupancy

To investigate mechanisms by which EHMT2 activates DKK1, we first carried out ChIP-seq analysis of EHMT2 occupancy in RD18 cells. EHMT2 enrichment was found mostly at gene promoters (*Figure 4A*) and its occupancy was apparent at the *DKK1* promoter (*Figure 4B*). To validate these results, ChIP-PCR was done in RD18 cells. A significant enrichment was seen indicating that EHMT2 directly binds to the *DKK1* promoter (*Figure 4C*). To further ascertain the specificity of EHMT2 occupancy at the *DKK1* promoter, we performed ChIP-PCR at chromatin regions before and after the EHMT2 peak at the promoter. Neither region showed significant EHMT2 enrichment (*Figure 4D and E*). Homer analysis of the ChIP-seq data for DNA motif enrichment at predicted EHMT2 binding sites revealed KLF7 as one of the top DNA motifs associated with EHMT2-predicted binding sites (*Figure 4—figure supplement 1A*). Sp1 is a member of the Krüppel-like factors (KLFs) all of which share a highly conserved DNA binding domain with high sequence similarity. The transcription factor Sp1 and the co-activator p300 have previously been shown to regulate *DKK1* expression (*Peng et al., 2017*; *Polakowski et al., 2010*). Consistent with these studies, Sp1 occupancy was detected at the *DKK1* promoter (*Figure 4F*) in RD18 cells. Intriguingly, both Sp1 and p300 enrichment were decreased upon treatment with UNC0642 compared to control RD18 cells. Correspondingly, a reduction in H3K9ac, a mark of transcriptional activation, was apparent in RD18 cells (*Figure 4G–I*). A decrease in p300 and H3K9ac occupancy was also observed in shEHMT2 RD cells (*Figure 4—figure supplement 1B and C*).

Sp1 interacts with p300 through its DNA binding domain (*Suzuki et al., 2000*). We therefore examined if Sp1 and p300 interaction was altered by UNC0642 by proximity ligation assay (PLA). The interaction between Sp1 and p300 decreased upon UNC0642 treatment in both RD18 and RD cell lines, as well as in HSMM (*Figure 4J* and *Figure 4—figure supplement 1D and E*). However, EHMT2-Sp1 and EHMT2-p300 interaction remained unchanged (*Figure 4—figure supplement 1F and G*). PLA for single antibody controls with p300, EHMT2, and Sp1 antibodies in RD 18 cells showed minimal background signals (*Figure 4—figure supplement 1H*). To validate the PLA data, we examined Sp1–p300 interaction under UNC0642 treatment in RD cells by immunoprecipitation assays. Immunoprecipitation with anti-Sp1 antibody confirmed a decrease in p300 association upon UNC0642 treatment (*Figure 4K*). Thus, our results indicate that EHMT2 binding at the promoter results in increased Sp1 and p300 occupancy and active transcription of *DKK1*.

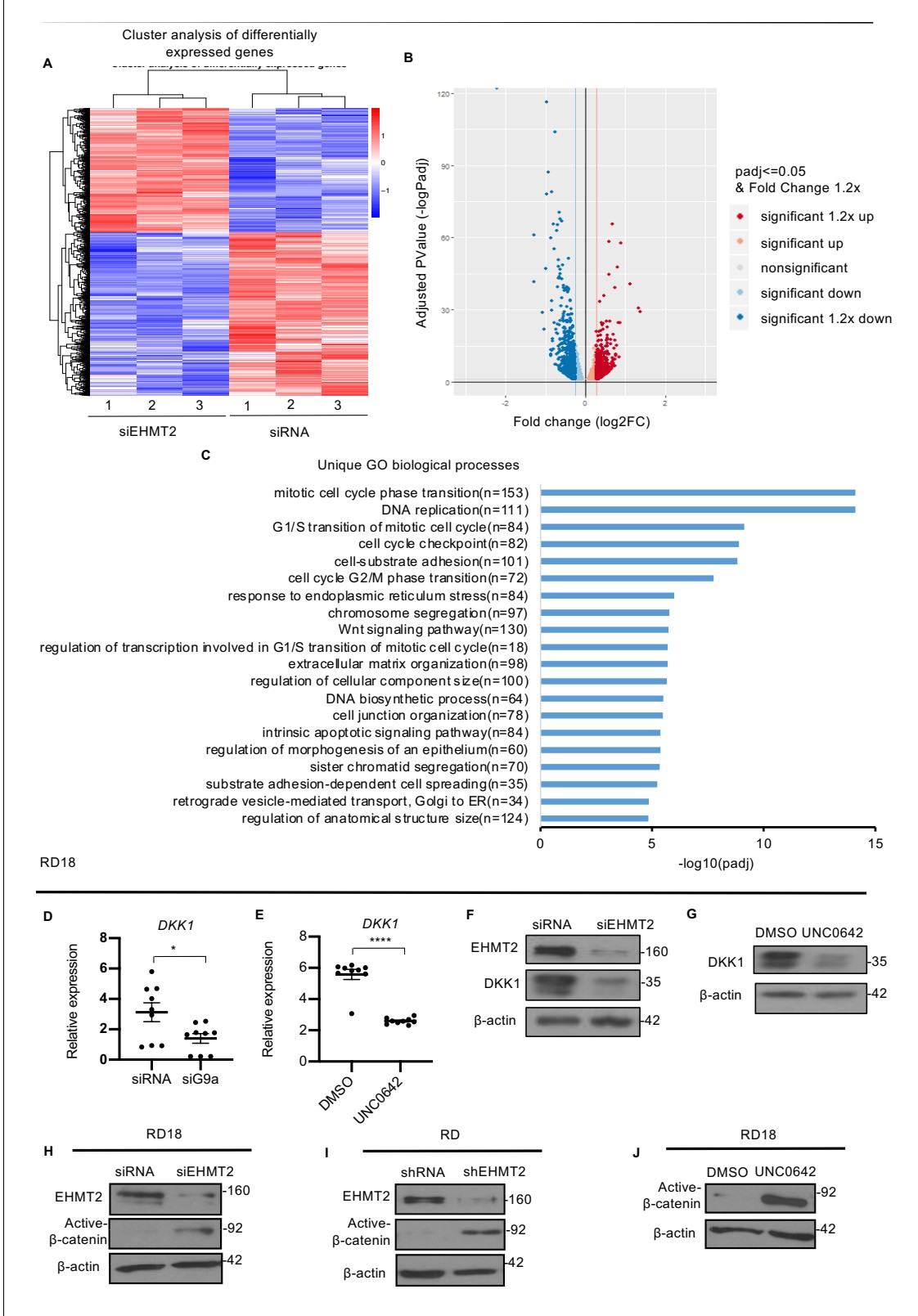

**Figure 3.** EHMT2 regulates DKK1 and Wnt signaling. (**A**) RNA-seq heatmap showing hierarchical clustering of differentially expressed genes. RNA-Seq was performed with control and siEHMT2 RD cells (n = 3). Red represents high expression and blue represents low expression. (**B**) Volcano plot showing distribution of differentially expressed genes upon EHMT2 knockdown in RD cells. (**C**) GO enrichment histogram displaying top 20 unique significantly enriched biological processes upon EHMT2 knockdown in RD cells based on *p*-adjusted value where n signifies the number of differentially

*Figure 3 continued on next page*

*Figure 3 continued*

expressed genes concerning the GO term. (**D** and **E**) qPCR analysis for *DKK1* mRNA in RD18 control and siEHMT2 cells and upon 2.5 µM of UNC0642 treatment (n = 3). Values correspond to the average ± SEM. (**F** and **G**) DKK1 protein was analyzed in control and siEHMT2 RD18 cells and in DMSO and 2.5 µM of UNC0642-treated RD18 cells. Representative images of three different experiments are shown. (**H–J**) Western blot analysis showed increased active-β-catenin in siEHMT2 RD18 cells relative to controls, in stable RD shEHMT2 cells, and upon UNC0642 treatment in RD18 cells as indicated. Representative images from three different experiments are shown. In (**D** and **E**) data from three independent biological replicates each with three technical replicates were plotted. Statistical significance was calculated by unpaired two-tailed *t*-test. *p≤0.05, ***p≤0.001.

The online version of this article includes the following source data and figure supplement(s) for figure 3:

**Source data 1.** qPCR data for DKK1 expression in RD18 cells upon G9a knockdown.
**Source data 2.** qPCR data for DKK1 expression in RD18 cells upon G9a activity inhibition by UNC0642.
**Figure supplement 1.** Validation of RNA-sequencing analysis.

## EHMT2 inhibits differentiation and promotes proliferation through DKK1-mediated antagonism of Wnt signaling

Canonical Wnt signaling induces myogenic differentiation and decreases proliferation (*Suzuki et al., 2015*; *Brack et al., 2008*). As DKK1 is a well-characterized inhibitor of canonical Wnt signaling, we examined if the effect of EHMT2 is mediated by DKK1. Correlating with high endogenous EHMT2 expression, DKK1 was also overexpressed in all three lines compared to HSMM at both mRNA and protein levels (*Figure 5A and B*). Similar to *EHMT2* knockdown, *DKK1* knockdown in RD18 cells resulted in an increase in active-ß-catenin levels indicating an upregulation of canonical Wnt signaling (*Figure 5C*). Moreover, analogous to *EHMT2* knockdown, *DKK1* knockdown in RD18 cells resulted in a significant decrease in BrdU⁺ cells compared to control cells (*Figure 5D*). A corresponding increase in differentiation was also apparent by elevated MHC levels and *MYOG* expression in RD18 cells (*Figure 5E*). A similar decrease in BrdU⁺ cells and increase in MHC levels upon DKK1 knockdown were observed in RD cells (*Figure 5—figure supplement 1A and B*). To determine whether DKK1 mediates the effects of EHMT2, we performed rescue experiments in RD18 cell line. Recombinant DKK1 (rDKK1) was added to siEHMT2 cells for 24 hr that resulted in the reduction of active β-catenin seen in siEHMT2 cells. Interestingly, in the presence of rDKK1, the increase in MHC⁺ cells and *MYOG* expression in siEHMT2 cells was reversed to control levels. Similarly, the decrease in BrdU⁺ cells upon EHMT2 knockdown were restored to levels comparable to control (*Figure 5F*). To further validate that EHMT2 mediates its effects on canonical Wnt signaling, we used another Wnt antagonist, a porcupine inhibitor, LGK974. Similar to rDKK1, LGK974 reversed the effects of EHMT2 knockdown on proliferation, differentiation, and active β-catenin levels (*Figure 5G*) indicating that EHMT2 mediates the differentiation block by activating *DKK1* expression that in turn suppresses canonical Wnt signaling.

In order to examine the effect of EHMT2 in regulating DKK1 and Wnt signaling in vivo, we injected RD cells in BALB/c nude mice. Once the tumors were palpable, mice were injected intraperitoneally every 2 days with UNC0642 or with control vehicle. Treatment with UNC0642 resulted in reduced tumor growth compared to the control group without any significant changes in body weight (*Figure 6A*). By immunohistochemical analysis (*Figure 6B*) we confirmed a decrease in H3K9me2 in tumors from mice treated with UNC0642 indicating efficacy of the drug in vivo. The proliferation marker Ki67 was decreased, whereas MHC⁺ cells were increased in tumors from mice treated with UNC0642. Moreover, DKK1 was decreased upon UNC0642 treatment, and correspondingly active-ß-catenin levels were elevated (*Figure 6B*).

To verify that the effects of EHMT2 are mediated via an impact on Wnt signaling in vivo, we next injected control shRNA and shEHMT2 cells. Once the tumors were palpable, mice injected with shEHMT2 cells were treated with LGK974 or treated with vehicle alone every alternate day. The shRNA control group was also injected with the vehicle (*Figure 6C*). Tumor volume in mice injected with shEHMT2 cells was reduced compared to the control group. However, mice injected with shEHMT2 cells and treated with LGK974 showed tumor volumes comparable to the control group. Body weight of mice did not show any significant changes over the course of this treatment. We analyzed two tumors from each cohort by immunohistochemistry (IHC). As expected, EHMT2 expression was decreased in mice injected with shEHMT2 cells that correlated with decrease in Ki67⁺ cells and increase in MHC⁺ cells compared to tumors of the control group. This alteration was however not

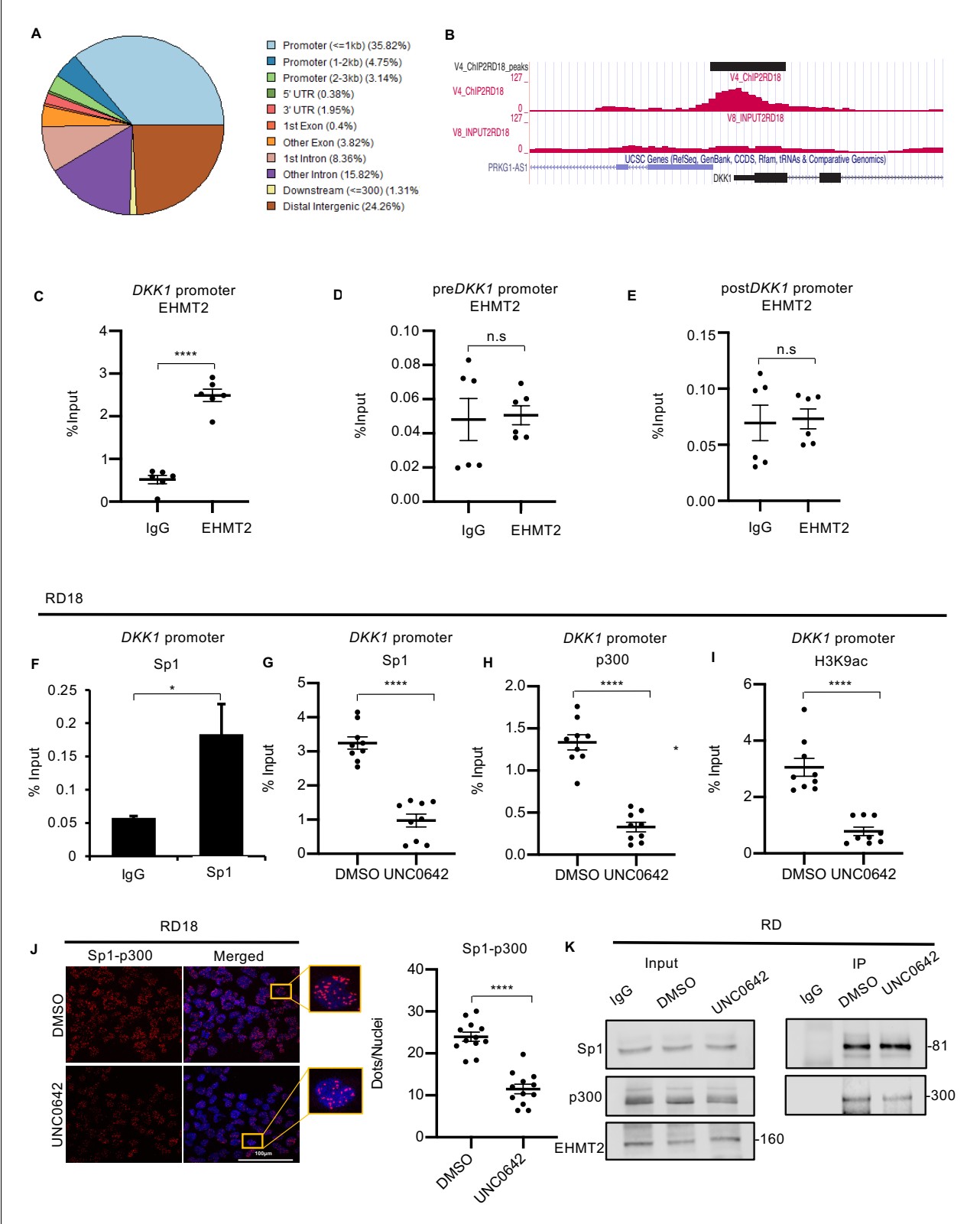

**Figure 4.** EHMT2 binds to the *DKK1* promoter and regulates Sp1/p300 occupancy in a methyltransferase activity-dependent manner. (**A**) ChIP-seq analysis in RD18 cells showed EHMT2 occupancy at different regions of the chromatin. (**B**) Snapshot of EHMT2 binding peak at the *DKK1* promoter from the UCSC genome browser. (**C**) EHMT2 occupancy at the *DKK1* promoter was validated by ChIP-PCR (n = 3) in RD18 cells. (**D** and **E**) The specificity of the EHMT2 occupancy was validated by ChIP-PCR using primers spanning the chromatin region of enrichment, before (preDKK1) and after

*Figure 4 continued on next page*

_Figure 4 continued_

(postDKK1) the EHMT2 peak at the DKK1 promoter (n = 3) in RD18 cells. The dot plot shows EHMT2 enrichment compared to IgG which was used as a control. Values correspond to the average ± SEM. (**F**) Sp1 occupancy was analyzed by ChIP-PCR at the _DKK1_ promoter in RD18 cells. IgG was used as a control (n = 2). Bar graph for one representative biological experiment with three technical replicates is shown. Values correspond to the average ± SD. (**G–I**) Sp1, p300, and H3K9ac enrichment at the _DKK1_ promoter was analyzed in 2.5 µM of UNC0642-treated RD18 cells compared to DMSO controls. The dot plots show reduced enrichment in UNC0642-treated cells (n = 3). Values correspond to the average ± SEM. (**J**) Proximity ligation assay was done to examine Sp1 and p300 teraction in control and 2.5 µM of UNC0642-treated RD18 cells. Images were captured using confocal microscopy. The dot plot shows the number of dots per nuclei in UNC0642-treated cells compared to control cells (n = 3). Each dot represents an interaction. Values correspond to the average ± SEM. (**K**) Immunoprecipitation with anti-Sp1 antibody was done to examine interaction with p300 in control and 2.5 µM of UNC0642-treated RD cells. 10% lysate was run as input and immunoblotted for Sp1, p300, and EHMT2 by western blotting. The numbers indicate molecular weight of proteins. In (**C–E**) and (**G–J**) data from three independent biological replicates each with three technical replicates were plotted. Statistical significance was calculated by unpaired two-tailed _t_-test. *p≤0.05, **p≤0.01, ***p≤0.001, ****p≤0.0001.

The online version of this article includes the following source data and figure supplement(s) for figure 4:

**Source data 1.** ChIP qPCR data for G9a occupancy on DKK1 promoter, pre DKK1 promoter region and post DKK1 promoter region in RD18 cells.
**Source data 2.** ChIP qPCR data for Sp1 occupancy on DKK1 promoter in RD18 cells.
**Source data 3.** ChIP qPCR data for Sp1, p300 and H3K9ac occupancy on DKK1 promoter upon G9a activity inhibition by UNC0642.
**Source data 4.** PLA quantification data of Sp1-p300 interaction in RD18 cells upon G9a activity inhibition by UNC0642.
**Figure supplement 1.** EHMT2 regulates Sp1 and p300 occupancy at the _DKK1_ promoter.

seen in the tumors from mice injected with shEHMT2 cells that underwent LGK974 treatment. Consistently, the reduction in DKK1 and increase in active-ß-catenin in tumors from shEHMT2 injected mice were reversed in upon LGK974 treatment (_Figure 6D_).

Taken together, our studies demonstrate that inhibition of EHMT2 expression or activity promotes differentiation and reduces tumor progression by regulating Wnt signaling. Integration of RNA-Seq and ChIP-Seq data (_Figure 5—figure supplement 1C_) showed many genes involved in myogenic differentiation, cell cycle progression, and metabolic pathways (_Figure 5—figure supplement 1D–F_) to be directly or indirectly regulated by EHMT2. These genes, independent of Wnt signaling, may contribute to oncogenic effects of EHMT2 in ERMS cells.

## Discussion

In this study, we uncovered an EHMT2-dependent epigenetic node that results in repression of Wnt signaling in ERMS. We propose that by activating _DKK1_ expression, EHMT2 maintains Wnt signaling in a repressed state, and thus prevents the transition of myoblasts to a differentiated state. These findings underscore specific epigenetic mechanisms to reactivate Wnt signaling and induce differentiation in ERMS.

There has been a resurgence of interest in differentiation therapy as a viable treatment option for solid tumors (_Cruz and Matushansky, 2012_). The Wnt, Notch, and Hedgehog pathways play a pivotal role in balancing proliferation, self-renewal, and differentiation during embryonic myogenesis. Not surprisingly, deregulation of these developmental pathways has been reported in ERMS (_Hatley et al., 2012_; _Chen et al., 2014_; _Satheesha et al., 2016_; _Ignatius et al., 2017_) and consequently, drugs targeting each of these pathways are being tested. However, current inhibitors do not selectively target specific pathways and have either unacceptable toxicity, or do not show marked clinical improvement. Several studies have demonstrated suppression of Wnt signaling in RMS cells, which mostly do not stain positively for nuclear β-catenin. Also, no mutations in the β catenin gene have been reported (_Bouron-Dal Soglio et al., 2009_; _Annavarapu et al., 2013_). The suppression of Wnt signaling is a critical contributor to the differentiation block in ERMS, and induction of the pathway leads to cell cycle exit and differentiation (_Singh et al., 2010_; _Chen et al., 2014_). GSK3ß inhibitors are the most prominent Wnt signaling activators and showed promising effects in inducing differentiation in a zebrafish model of ERMS (_Chen et al., 2014_). GSK3ß is a constitutive serine/threonine protein kinase that inhibits canonical Wnt signaling by phosphorylating ß-catenin and triggering its degradation in the cytoplasm (_Wu and Pan, 2010_). However, GSK3ß is involved in many pathways with more predicted substrates than any known kinase. Thus, its inhibition is not a

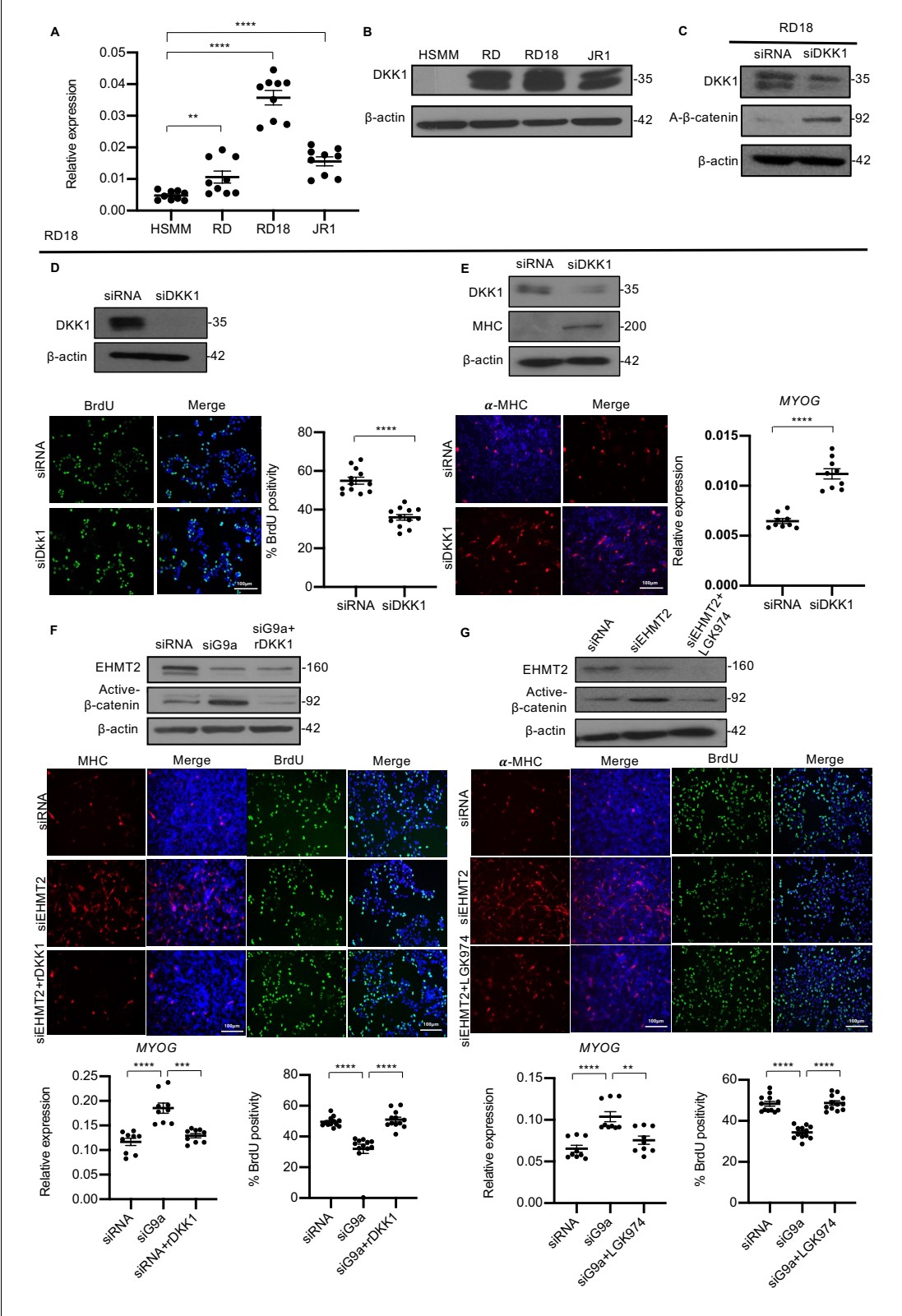

**Figure 5.** DKK1 is a downstream effector of EHMT2 function. (**A**) *DKK1* mRNA was examined by qPCR (n = 3) in human skeletal muscle myoblast (HSMM), RD, RD18, and JR1 cells. Values correspond to the average ± SEM. (**B**) DKK1 protein levels were analyzed by western blotting in HSMM, RD, RD18, and JR1. A representative image from three different experiments is shown. (**C**) Effect on canonical Wnt signaling upon knockdown of DKK1 was examined by analyzing active-β-catenin protein levels in control and siDKK1 RD18 cells. (**D**) DKK1 knockdown was analyzed in control and

*Figure 5 continued on next page*

*Figure 5 continued*

siDKK1 RD18 cells by western blot. Proliferation was analyzed in RD18 control and siDKK1 cells (n = 3) with anti-BrdU antibody. The dot plot shows the percentage of BrdU$^+$ cells. Values correspond to the average ± SEM. (E) Differentiation was analyzed in control and siDKK1 RD18 cells that were cultured for 5 days in DM. Cells were analyzed by western blot and immunofluorescence and using anti-MHC antibody as indicated. A representative image of three different experiments is shown. *MYOG* expression was analyzed by qPCR (n = 3) at day 2 of differentiation. Values correspond to the average ± SEM. (F) Control, siEHMT2 cells, and siEHMT2 RD18 cells treated with rDKK1 for 24 hr and tested for active-β-catenin levels. Differentiation and proliferation were analyzed (lower panels) by MHC$^+$ cells and BrdU$^+$ cells as indicated. Representative images of three different experiments are shown. *MYOG* expression was analyzed by qPCR (n = 3) and the percentage of BrdU$^+$ cells is shown in the dot plots. Values correspond to the average ± SEM. (G) Western blot showing active-β-catenin levels in control, siEHMT2 cells, and siEHMT2 RD18 cells treated with LGK974 for 24 hr. A representative image of three different experiments is shown. MHC$^+$ and BrdU$^+$ cells were analyzed. A representative image of three different experiments is shown. *MYOG* expression in control, siEHMT2, and siEHMT2 RD18 cells treated with LGK974 was analyzed by qPCR (n = 3). Values correspond to the average ± SEM. Statistical significance in (A) and (D–G) was calculated by unpaired two-tailed *t*-test. **p≤0.01, ***p≤0.001, ****p≤0.0001.

The online version of this article includes the following source data and figure supplement(s) for figure 5:

**Source data 1.** qPCR data for endogenous DKK1 expression in ERMS cell lines.
**Source data 2.** BrdU quantification data in RD18 cells upon DKK1 knockdown.
**Source data 3.** qPCR data for day 2 myogenin expression in RD18 cells upon DKK1 knockdown.
**Source data 4.** qPCR data for day 2 myogenin expression in RD18 cells upon rDKK1 treatment in G9a knockdown cells.
**Source data 5.** BrdU quantification data in RD18 cells upon rDKK1 treatment in G9a knockdown cells.
**Source data 6.** qPCR data for day 2 myogenin expression in RD18 cells upon LGK974 treatment in G9a knockdown cells.
**Source data 7.** BrdU quantification data in RD18 cells upon LGK974 treatment in G9a knockdown cells.
**Figure supplement 1.** Effect of DKK1 knockdown in RD cells and integration of RNA-seq and ChIP-seq data.

specific strategy to induce Wnt signaling. This is also emphasized by slow progress of existing GSK3ß inhibitors toward clinical translation (*Pandey and DeGrado, 2016*). Tideglusib, an irreversible GSK3ß inhibitor, was recently tested against RMS PDX models, where the highest safe dosage failed to both induce myogenic differentiation and affect cancer progression in PDX models (*Bharathy et al., 2017*). Consequently, development of alternative molecularly targeted therapies that induce Wnt signaling is a critical goal in this disease.

Our data demonstrates an epigenetic mechanism to activate Wnt signaling and overcome the differentiation block in ERMS. Interestingly EHMT2 activates *DKK1* in a methylation-dependent manner resulting in the suppression of Wnt signaling in ERMS. While methylation-dependent silencing or repression by EHMT2 has been described, only a few studies have demonstrated methylation-dependent activation of gene expression. EHMT2 occupancy at its promoter demonstrates that it directly regulates *DKK1* expression. Consistent with previous studies, we found Sp1 binding at the *DKK1* promoter and inhibition of EHMT2 led to its decreased occupancy along with the coactivator p300. The hypothesis that G9a activation of *DKK1* is mediated by H3K9me2 loss needs to be tested further and the mechanisms by which G9a activity regulates Sp1–p300 occupancy at the promoter need further investigation. Interestingly, RNA-seq analysis revealed de-regulation of several Sp1 target genes such as *IGBP2*, *IGBP3*, *HB-EGF*, *FGFR1*, *CCND1*, *VEGFA*, and *VEGFC* suggesting that the effect of EHMT2 inhibition on Sp1 transcriptional activity might not be limited to *DKK1*.

Pan-HDAC and pan-DNMT inhibitors have been explored in ERMS (*Vleeshouwer-Neumann et al., 2015*; *Gnyszka et al., 2013*). HDAC inhibitors often exert effects independent of their epigenetic roles (*Vleeshouwer-Neumann et al., 2015*) and application of DNMT inhibitors (*Gnyszka et al., 2013*) is restricted due to their toxicity in healthy cells. EZH2 inhibitors are the only other specific epigenetic inhibitors that demonstrate a strong phenotype in ERMS (*Ciarapica et al., 2014*). In this study, a drug screen of 15 methyltransferase inhibitors in two different ERMS cell lines showed that small molecule inhibitors targeting EHMT2 activity are very effective. Thus, our data support targeting EHMT2 as a therapeutic approach in ERMS, particularly since its deletion does not impact development of muscle (*Zhang et al., 2016*). We have recently shown that EHMT2 is deregulated in ARMS as well (*Bhat et al., 2019*). Together with the herein described role of EHMT2 in ERMS, these observations clearly suggest the importance of differentiation in these therapies and imply common features between these two subtypes of RMS.

# Materials and methods

## Key resources table

| Reagent type (species) or resource | Designation | Source or reference | Identifiers | Additional information |
|---|---|---|---|---|
| Strain, strain background (*Mus musculus* female) | Nude mice | In Vivos | *C.Cg/AnNTac-Foxn1^{nu}NE9 BALB/c* RRID:IMSR_TAC:balbnu | BALB/c inbred model |
| Cell line (*Homo sapiens*) | HSMM | Lonza Inc | #: CC-2580 | Isolated from upper arm or leg muscle tissue of normal donors and sold at second passage |
| Cell line (*H. sapiens*) | RD18 | Peter Houghton and Rosella Rota | RRID:CVCL_IU87 | Clone cells derived from RD cells |
| Cell line (*H. sapiens*) | RD | Peter Houghton and Rosella Rota | RRID:CVCL_1649 | Patient-derived cell line from pelvic mass of 7-year-old female |
| Cell line (*H. sapiens*) | JR1 | Peter Houghton and Rosella Rota | RRID:CVCL_J063 | Patient-derived cell line from lung metastasis of 7-year-old female |
| Cell line (*H. sapiens*) | RD shscm | This study | | Transfected with shRNA control lentivirus particles |
| Cell line (*H. sapiens*) | RD shEHMT2 | This study | | Transfected with shEHMT2 lentivirus particles |
| Transfected construct (*H. sapiens*) | SmartPool non-targeting siRNA | Dharmacon | D-001810-10-20 | Negative control of four siRNAs designed for minimal targeting |
| Transfected construct (*H. sapiens*) | SmartPool siRNA against EHMT2 | Dharmacon | L-006937-00-0010 | A mixture of four siRNA provided as a single reagent |
| Transfected construct (*H. sapiens*) | SmartPool siRNA against DKK1 | Dharmacon | L-003843-01-0010 | A mixture of four siRNA provided as a single reagent |
| Transfected construct (*H. sapiens*) | shRNA control lentivirus particles | Santa Cruz Biotechnology Inc | sc-108080 | Negative control. 200 µl viral stock containing $1 \times 10^6$ IFU. Encodes nonspecific scrambled shRNA |

*Continued on next page*

*Continued*

| Reagent type (species) or resource | Designation | Source or reference | Identifiers | Additional information |
|---|---|---|---|---|
| Transfected construct (*H. sapiens*) | shEHMT2 control lentivirus particles | Santa Cruz Biotechnology Inc | sc-43–777V | 200 µl of viral stock containing $1 \times 10^6$ IFU. Pools of three to five target-specific sequences |
| Antibody | Rabbit monoclonal EHMT2 | Cell Signalling | #3306S | 1:300, western blot |
| Antibody | Mouse monoclonal MHC | Santa Cruz Biotechnology | Sc-32732 | 1:300, western blot |
| Antibody | Mouse monoclonal Myogenin | Santa Cruz Biotechnology | Sc-12732 | 1:250, western blot |
| Antibody | Mouse monoclonal Dkk1 | Santa Cruz Biotechnology | Sc-374574 | 1:300, western blot, 1:200 for IHC |
| Antibody | Mouse monoclonal active-ß-catenin | Merk Millipore | 05–665 | 1:500, western blot, 1:300 for IHC |
| Antibody | Rabbit polyclonal H3K9me2 | Cell Signaling | 9753S | 1:1000, western blot, 1:200 for IHC |
| Antibody | Mouse monoclonal ß-actin | Sigma-Aldrich | A2228 | 1:10,000, western blot |
| Antibody | Rabbit polyclonal H3 | Abcam | Ab-1791 | 1:10,000, western blot |
| Antibody | Mouse monoclonal Sp1 | Santa Cruz | Sc-17824 | 1:50 for PLA |
| Antibody | Rabbit Polyclonal Sp1 Rabbit | Merck Millipore | 07–645 | 1:100 for PLA, 3 µg was used for ChIP. 2 µg was used for IP pull down. 1:500 dilution for immunoblotting |
| Antibody | Mouse monoclonal p300 | Abcam | Ab14984 | 1:1000 for PLA, 2 µg was used for ChIP. 1:500 dilution for immunoblotting |
| Antibody | Rabbit polyclonal H3K9ac | Abcam | Ab4441 | 2 µg was used for ChIP |
| Antibody | Rabbit polyclonal EHMT2 | Abcam | Ab40542 | 2 µg was used for ChIP, 1:200 for IHC |
| Antibody | Mouse monoclonal Ki67 | Leica Biosystems | PA0118 | 1:100 for IHC |

*Continued on next page*

*Continued*

| Reagent type (species) or resource | Designation | Source or reference | Identifiers | Additional information |
|---|---|---|---|---|
| Antibody | Mouse monoclonal MHC | Sigma Aldrich | M4276 | 1:200 for IHC and IF |
| Sequence-based reagent | Primers | This study | | As mentioned in Materials and methods |
| Peptide, recombinant protein | Human DKK1 | R and D Systems | 5439-dk-010 | 100 ng/ml, Sf21(baculovirus)-derived human DKK1 protein |
| Commercial assay or kit | PLA kit (Duolink in situ- fluorescence) | Sigma | DUO92101 | |
| Commercial assay or kit | Lipofectamine RNAiMax | Thermo Fisher Scientific | 13778150 | |
| Chemical compound | LGK974 | Selleck Chemicals | S7143 | 200 nM, porcupine inhibitor |
| Chemical compound | Polybrene | Sigma Aldrich | TR-1003 | 2 Ul of 8 mg/ml |
| Chemical compound | Puromycin dihydro chloride | Sigma Aldrich | P8833 | 1 µg/ml |
| Software, algorithm | GraphPad prism | | V9.0 | https://www.graphpad.com/ |

## Cell culture and drug sensitivity assays

RD, RD18, and JR1 ERMS cell lines were a kind gift from Peter Houghton (Nationwide Children's Hospital, Ohio, USA) and Rosella Rota (Bambino Gesu Children's Hospital, Rome, Italy). All cell lines were routinely tested and were negative for mycoplasma. RD18 and JR1 were cultured in RPMI 1640 with L-Glutamine (Thermo Fisher Scientific, Waltham, MA, USA) and 10% FBS (Hyclone, Logan UT, USA), whereas RD cells were cultured in Dulbecco's Modified Eagle Medium (DMEM) (Sigma, St Louis, MO, USA) with 10% FBS (Hyclone, Logan UT, USA). Primary HSMMs were purchased from Lonza Inc (Basel, Switzerland) and cultured in growth medium (SkGM-2 BulletKit). For transient knockdown, cells were transfected with 50 nM of human EHMT2-specific siRNA or human DKK1-specific siRNA (ON-TARGETplus siRNA SMARTpool, Dharmacon, Lafayette, CO, USA) containing a pool of three to five 19–25 nucleotide siRNAs. Control cells were transfected with 50 nM scrambled siRNA (ON-TARGETplus, non-targeting pool, Dharmacon) using Lipofectamine RNAiMax (Thermo Fisher scientific). Cells were analyzed 48 hr post-transfection for all assays. Knockdown efficiency was monitored by western blot. For generating stable knockdown cell lines, RD cells at 40–50% confluency were transduced with shRNA control lentivirus particles (Santa Cruz Biotechnology Inc), or shEHMT2 lentivirus particles (Santa Cruz Biotechnology Inc) and 2 µl polybrene (8 mg/ml) (Sigma-Aldrich) in DMEM basal medium. Six hours post-transduction, cell supernatants were replaced with DMEM medium (10% FBS) for 24 hr. Transduced cells were selected with 1 µg/ml puromycin (Sigma-Aldrich) for 4 days. For rescue experiments, siEHMT2 cells were treated with 100 ng/ml of rDKK1 (R and D Systems) or 200 nM of porcupine inhibitor LGK974 (Selleck Chemicals, Houston, USA). Both rDKK1 and LGK974 were added to the media 24 hr after transfection. For drug screening, RD and JR1 cells were seeded in 384 well plates at 200 cells per well together with methyltransferase inhibitors at 3, 1, and 0.3 µM. During the treatment, the media and the drugs were not replaced. The viability read-out was obtained at day 8 by MTS assay (as per manufacturer's instructions), and calculated as the ratio over control cells treated with an equivalent dilution of DMSO. The data are presented as a heatmap where red indicates viability ratio >control; white = control; and blue is less than control. The experiment was conducted in triplicate and (+)-JQ1 was used as a positive control. To determine the effect of UNC0642, HSMM and RD cells were treated with DMSO or UNC0642 for

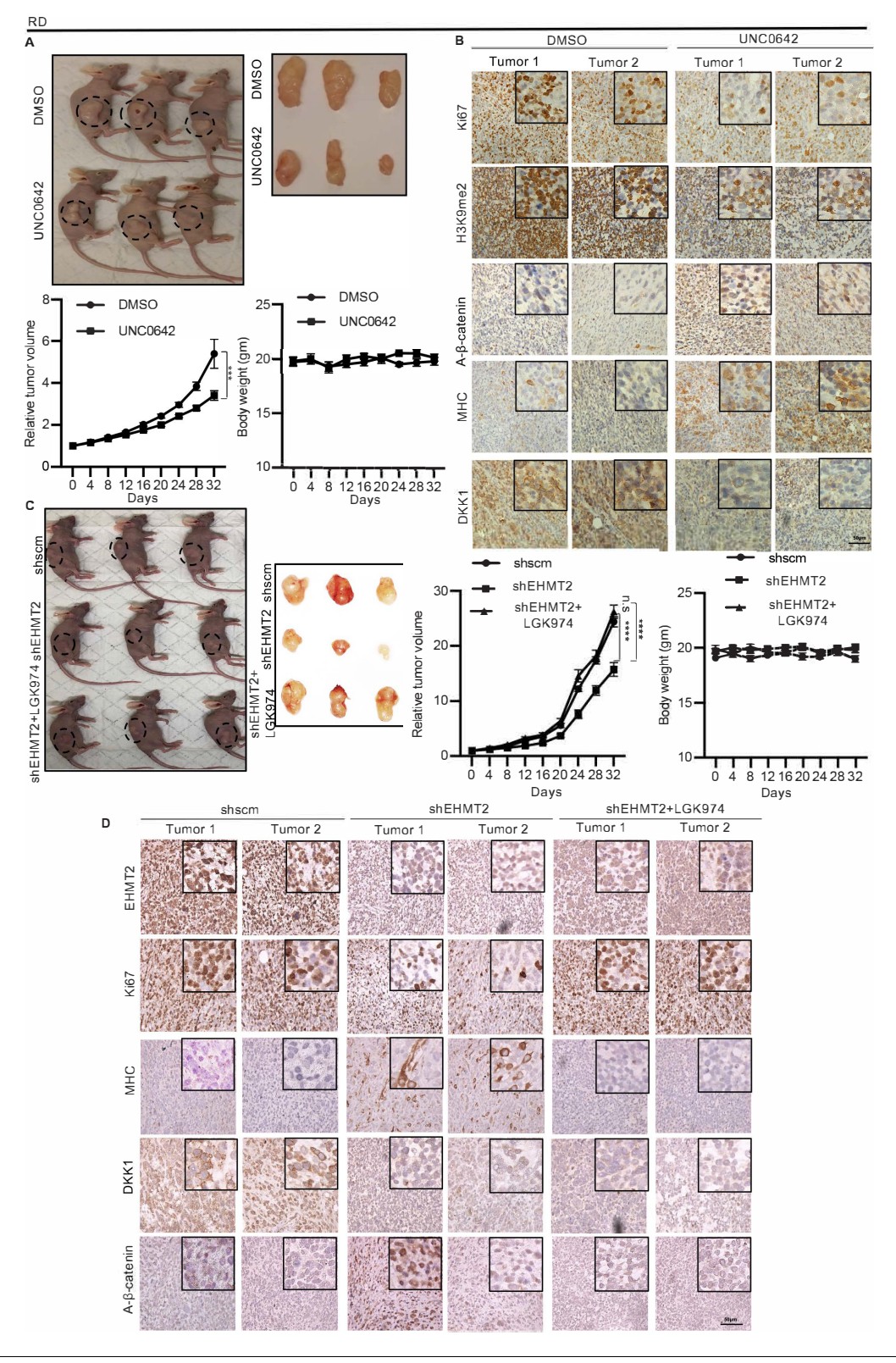

**Figure 6.** EHMT2 regulates tumor growth by regulation of DKK1 and Wnt signaling. (**A**) Nude mice were injected with RD cells. Once tumors were palpable, mice were treated with DMSO (n = 10) or UNC0642 (n = 10). Representative images of three mice in each group (left panel) and resected tumors (right panel) are shown. The relative tumor volume in UNC0642-treated group showed a significant decrease compared to controls although the body weight of mice did not show any significant change. Statistical significance was calculated using repeated-measure two-way ANOVA where

*Figure 6 continued on next page*

*Figure 6 continued*

***p$\leq$0.001. Values correspond to the average ± SEM. (B) Tumors from two control and two UNC0642-treated mice were analyzed by immunohistochemistry (IHC) using anti-Ki67, anti-H3K9me2, anti-active-ß-catenin, anti-MHC, and anti-DKK1 antibodies. Scale bar: 50 µm. Inset shows zoomed in images. (C and D) Mice were injected with shRNA RD cells (shscm) (n = 10) or shEHMT2 RD cells (n = 20). Once tumors were palpable, half of shEHMT2 injected mice were treated with vehicle and the rest with LGK974. (C) Representative images of mice (left panel) injected with shscm control cells, shEHMT2 cells, and shEHMT2 cells + LGK974 are shown. Representative images of the tumors (right panel) isolated from the three cohorts are shown. The relative tumor volume and the body weight of mice in each group were determined. Statistical significance was calculated using repeated-measure two-way ANOVA where ****p$\leq$0.0001. Values correspond to the average ± SEM. (D) Tumors from two different mice in each group were analyzed by IHC for Ki67, H3K9me2, DKK1, MHC, and active-ß-catenin staining as described above. Scale bar: 50 µm. Inset shows zoomed in images.

The online version of this article includes the following source data for figure 6:

**Source data 1.** Relative tumor volume and body weight of mice upon G9a activity inhibition by UNC0642.
**Source data 2.** Relative tumor volume and body weight of mice upon G9a knockdown and treatment of G9a knockdown tumors with LGK974.

6 days and counted with trypan blue. Media and drugs were replenished every 2 days. The experiments were conducted in technical duplicates in two independent biological replicates. IC50 was determined by culturing cells with 5000 nM, 3000 nM, 1000 nM, 500 nM, 250 nM, and 0 nM EHMT2 inhibitors for 6 days. The concentration of drug that affects 50% viability of the cells was determined using the CompuSyn software.

## Colony formation assay

2000 cells were seeded per well in a six-well plate. Cells were treated with 2.5 µM UNC0642 or equal volumes of DMSO. The treatment was carried for 9 days at the end of which cells were stained with crystal violet to visualize the colonies formed.

## Proliferation and differentiation assays

Proliferative capacity of cells was analyzed using 5-bromo-2'-deoxy-uridine (BrdU) labeling (Roche, Basel, Switzerland). Cells seeded on coverslips were pulsed with 10 µM BrdU for 60 min at 37°C. Cells were fixed with 70% ethanol at $-20$°C for 20 min and incubated with anti-BrdU antibody (1:100) for 60 min followed by anti-mouse Ig-fluorescein antibody (1:200) for 60 min. After mounting onto slides with DAPI (Vectashield, Vector Laboratories, CA, USA), images were captured using fluorescence microscope BX53 (Olympus Corporation, Shinjuku, Tokyo, Japan). For differentiation assays, RD, RD18, and JR1 cells were cultured for 2–5 days in DMEM supplemented with 2% horse serum (Gibco, Carlsbad, CA, USA) at 90–95% confluency. Differentiation was assessed by MHC staining. Cells were fixed with 4% paraformaldehyde for 20 min at room temperature (RT). Cells were blocked and permeabilized using 10% horse serum and 0.1% Triton X containing PBS. Cells were then incubated with anti-Myosin Heavy Chain (MHC) primary antibody (R and D Systems, Minneapolis, MN, USA) (1:400, 1 hr at RT) followed by 1 hr of 1:250 secondary goat anti-Mouse IgG (H+L) Highly Cross-Adsorbed Secondary Antibody, Alexa Fluor 568 (Thermo Fisher scientific). Coverslips were mounted with DAPI (Vectashield, Vector Laboratories, CA, USA) and imaged using upright fluorescence microscope BX53 (Olympus Corporation).

## Western blot analysis

Cells were lysed using RIPA or SDS lysis buffer supplemented with protease inhibitors (Complete Mini, Sigma-Aldrich). The following primary antibodies were used: anti-EHMT2 (#3306S, 1:300, Cell Signaling), anti-MHC (#sc-32732, 1:300, Santa Cruz Biotechnology), anti-Myogenin (#sc-12732, 1:250, Santa Cruz Biotechnology), anti-DKK1 (#sc374574, 1:300, Santa Cruz Biotechnology), anti-active-ß-catenin (#05–665, 1:500, Merck Millipore), anti-H3K9me2 (#9753S, 1:1000, Cell Signaling), anti-ß-actin (#A2228, 1:10,000; Sigma-Aldrich), and anti-H3 (#ab1791, 1:10,000; Abcam). Appropriate secondary antibodies (IgG-Fc Specific-Peroxidase) of mouse or rabbit origin (Sigma Aldrich) were used.

## Proximity Ligation Assay

PLA was performed using the Duolink in situ-fluorescence (Sigma DUO92101). For EHMT2 and Sp1 interaction, anti-EHMT2 (Cell Signaling, 1:50) and anti-Sp1(1:50 Santa Cruz) antibodies were used. For Sp1 and p300 interaction studies, Sp1 (Millipore, 1:100) and p300 (Abcam, 1:1000) antibodies were used. Images were captured under FluoView FV1000 confocal fluorescence microscope (Olympus) at 60× (oil). For quantifying PLA signals, particle analysis was performed using Fiji/ImageJ software, and pixel area size of 2–50 was assigned for calculating the total number of PLA signals per field. PLA signals as dots per nuclei were calculated for at least three microscopic fields.

## Transcriptome analysis and quantitative real-time polymerase chain reaction (qPCR)

For RNA sequencing analysis, RNA was isolated from control and siEHMT2 cells in triplicate using Trizol. RNA was sequenced using Illumina high-throughput sequencing platform. CASAVA base recognition was used to convert raw data file to Sequence Reads and stored in FASTQ(fq) format. Raw reads were then further filtered in order to achieve clean reads using the following filtering conditions: reads without adaptors, reads containing number of base that cannot be determined below 10%, and at least 50% bases of the reads having Qscore denoting Quality value $\leq 5$. For mapping of the reads STAR software was used to align the reads against hg19 *Homo sapiens* reference genome. 1M base was used as the sliding window for distribution of the mapped reads. For differential expression gene (DEG) analysis Readcount obtained from gene expression analysis was used. Differential expression significance analysis of two experimental groups was done by Novogene using the DESeq2 R package and an adjusted p-value of 0.05 was applied (padj <0.05). For analysis of the differentially expressed genes, Gene Ontology analysis was done using cluster Profiler (*Yu et al., 2012*) software for GO terms with corrected p-value less than 0.05.

For qPCR analysis, total RNA was extracted using Trizol (Thermo Fisher Scientific) and quantified using Nanodrop. Messenger RNA (mRNA) was converted to a single-stranded complementary DNA (cDNA) using iScript cDNA Synthesis Kit (Bio-Rad). qPCR was performed using Lightcycler 480 SYBR Green 1 Master Kit (Roche). PCR amplification was performed as follows: 95℃ for 5 min, followed by 95℃ for 10 s, annealing at 60℃ for 10 s, followed by 45 cycles at 72℃ for 10 s. Light Cycler 480 software (version 1.3.0.0705) was used for analysis. CT values of samples were normalized to internal control GAPDH to obtain delta CT (ΔCT). Relative expression was calculated by $2{-}\Delta CT$ equation. qPCR was done using reaction triplicate and at least two independent biological replicates were done for each analysis. Primer sequences for *EHMT2* are 5'-TGGGCCATGCCACAAAGTC-3' and 5'-CAGATGGAGGTGATTTTCCCG-3'; for *MYOG* are 5'-GCCTCCTGCAGTCCAGAGT-3' and 5'-AGTGCAGGTTGTGGGGCATCT-3', and for *DKK1* are 5'-CGGGAATTACTGCAAAAATGGA-3' and 5'-GCACAGTCTGATGACCGGAGA-3'.

## Chromatin immunoprecipitation (ChIP)

Chromatin immunoprecipitation-sequencing (ChIP-Seq) was done using 20 million RD18 cells and anti-EHMT2 antibody (Abcam, Cambridge, MA, USA) as described (*Bhat et al., 2019*). Sequencing reads were mapped against human reference genome hg19. High quality mapped reads (MAPQ $\geq$ 10) were retained and potential duplicates were removed using SAMtools. EHMT2 binding sites were predicted from the libraries using MACS2. Read density was computed in the format of bigwig using MEDIPS with 50 bp window width. The prediction revealed 48,999 binding sites overlapping with promoters, gene body, and intergenic regions. We used GENCODE v19 to define promoters (±2.5 kb from TSS) and gene body. Mid-point predicted binding sites were used in this analysis. Of the 48,999 binding sites 49% (n = 24,176) localized at promoters, 29% (n = 14,252) at gene bodies, and 21% (n = 10,571) at inter-genic regions. To demonstrate the read density around annotated TSS, we identified the promoters of the TSS overlapping with predicted EHMT2 binding sites. The TSS were then extended ±20 kb and the binding signal was computed in each window of size 100 bp. Average read density for each window was computed using bigWigAverageOverBed. GENECODEv19 was used to classify promoter-bound EHMT2 binding. Peaks were annotated using ChIPseeker (*Yu et al., 2015*) and ChIPpeakAnno (*Zhu et al., 2010*) against genes model of UCSC, hg19, knownGene (TxDb.Hsapiens.UCSC.hg19.knownGene). Differential expression count matrix was analyzed using R. Genes with an adjusted p-value less than 0.05 were labeled as significantly

differentially expressed. From that list, genes with an absolute fold change $\geq$ 1.2 were selected for further analysis. GO analysis of gene subsets was performed using Metascape (*Zhou et al., 2019*). The ChIP-Seq data are compliant with MIAME guidelines and have been deposited in the NCBI GEO database.

ChIP-PCR was done as previously described (*Bhat et al., 2019*). Relative enrichment was calculated using $2-\Delta CT$ equation. The following antibodies were used for ChIP assays: ChIP-grade anti-EHMT2 (Abcam), anti-H3K9ac (Abcam), Sp1 (Rabbit Millipore), and p300 (Abcam). Primers used for ChIP at the *DKK1* promoter were: Forward: 5'-CCGGATAATTCAACCCTTACTGCC-3' and Reverse: 5'-GGAGCATTCCGGCCCCTTGGGAG-3'; for chromosome region before EHMT2 occupancy at *DKK1* promoter region were Forward: 5'-ACCTCAAAGCCGGGGATCTA-3' and Reverse: 5'-CTC TAGCAAGACGCCTCTGA-3'; and for the region after the EHMT2 occupancy at the *DKK1* promoter were Forward: 5'-AACCCTTCCCACAGCCGTA-3' and Reverse: 5'-CGAGACAGATTTGCACGCC-3'.

## Immunoprecipitation (IP)

For immunoprecipitation, cells were lysed using NP40 buffer. 1 mg cell lysate was precleared and 2 μg of Sp1 antibody (rabbit, Millipore) was added for immunoprecipitation. Samples were loaded and run in SDS PAGE followed by immunoblotting with p300 antibody (Abcam). Ten percent lysate was run as input and immunoblotted with anti-Sp1, anti-p300, and anti-EHMT2 antibodies.

## Mouse xenograft experiments

Six-week-old C.Cg/AnNTac-Foxn1$^{nu}$NE9 female BALB/c nude mice (InVivos, Singapore) were injected subcutaneously in the right flank with control RD cells ($10 \times 10^6$). Once tumors were palpable, one group (n = 10/group) was injected intraperitoneally with vehicle (5% DMSO in PBS), and the other with UNC0642 (5 mg/kg body weight in 5% DMSO) every alternate day. Tumor diameter and volume were calculated as described (*Bhat et al., 2019*). Resected tumors were fixed and paraffin sections were immunostained with various antibodies. To determine the role of Wnt signaling, one group of mice were injected with RD shcontrol cells (n = 10/group) and two groups with RD shEHMT2 cells. Once tumors were palpable, the control group and one shEHMT2 group were injected intraperitoneally with control vehicle (2% DMSO in corn oil), and one group of shEHMT2 mice was injected with LGK974 (5 mg/kg body weight in 2% DMSO). Tumor growth and body weight were recorded as described (*Bhat et al., 2019*). All animal procedures were approved by the Institutional Animal Care and Use Committee.

## Immunohistochemistry

Paraffin sections from 16 primary ERMS archival tumor specimens and three normal muscles from National University Hospital (NUH) and KK Women's and Children Hospital in Singapore were analyzed by IHC using anti-EHMT2 antibody (1:50 dilution, Cell Signaling) as described (*Bhat et al., 2019*). Negative controls were performed using secondary antibody only. Images were captured with Olympus BX43 microscope (Ina-shi, Nagano, Japan). Approval was obtained from the ethics committee (IRB) at NUS. For IHC on mouse xenografts, sections were incubated with anti-EHMT2 (1:200, Abcam), anti-H3K9me2 (Abcam), anti-Ki67 (Leica Biosystems), anti-active-ß-catenin (1:300; Merck Millipore), anti-MHC (#M4276 1:200 Sigma-Aldrich), and anti-DKK1 (#ab61034 Abcam) antibodies followed by biotinylated goat anti-rabbit/anti-mouse IgG (H+L) secondary antibody (Vector Laboratories). Sections were washed and incubated with Vectastain Avidin–Biotin Complex (Vector Laboratories) for 20 min at 37°C.

## Statistical analysis

For statistical analysis, two-tailed non-parametric unpaired *t*-test was used to evaluate significance with the use of GraphPad prism 9.0 software. Each experiment had three biological replicates. Standard error of mean (SEM) was calculated for all data sets and a p-value less than 0.05 was considered statistically significant. *$p \leq 0.05$, **$p \leq 0.01$, ***$p \leq 0.001$, ****$p \leq 0.0001$. For in vivo experiments repeat-measure two-way ANOVA was used to calculate the statistical significance between different groups.

## Acknowledgements

We thank Peter Houghton and Rosella Rota for ERMS cell lines; Ooi Wen Fong, Genome Institute of Singapore for bioinformatics analysis; Philip Kaldis for CDK1 antibody, and David Virshup Duke-NUS for helpful discussions. We thank the Structural Genomics Consortium (SGC) for the epigenetics probe set. This work was supported by a National Medical Research Council grant (NMRC/OFIRG/0073/2018) to RT. AP was supported by the President's Graduate Scholarship at the National University of Singapore.

## Additional information

### Funding

| Funder | Grant reference number | Author |
|---|---|---|
| National Medical Research Council | NMRC/OFIRG/0073/2018 | Reshma Taneja |
| National University of Singapore | President's Graduate Scholarship | Ananya Pal |

The funders had no role in study design, data collection and interpretation, or the decision to submit the work for publication.

### Author contributions

Ananya Pal, Conceptualization, Formal analysis, Investigation, Visualization, Methodology, Writing - original draft, Writing - review and editing; Jia Yu Leung, Formal analysis, Validation, Investigation, Visualization, Methodology; Gareth Chin Khye Ang, Validation, Investigation, Visualization; Vinay Kumar Rao, Data curation, Validation, Investigation, Methodology; Luca Pignata, Resources, Formal analysis, Investigation, Methodology, Writing - original draft; Huey Jin Lim, Investigation, Visualization, Methodology; Maxime Hebrard, Data curation, Software; Kenneth TE Chang, Resources; Victor KM Lee, Resources, Visualization, Methodology; Ernesto Guccione, Resources, Data curation, Software, Funding acquisition, Writing - original draft, Writing - review and editing; Reshma Taneja, Conceptualization, Formal analysis, Supervision, Funding acquisition, Writing - original draft, Project administration, Writing - review and editing

### Author ORCIDs

Jia Yu Leung https://orcid.org/0000-0003-1317-8396
Kenneth TE Chang http://orcid.org/0000-0001-5244-4285
Reshma Taneja https://orcid.org/0000-0001-6214-6177

### Ethics

Animal experimentation: All animal procedures used in this study were approved by the Institutional Animal Care and Use Committee (IACUC) at the National University of Singapore under the protocol # R18-0208.

### Decision letter and Author response

Decision letter https://doi.org/10.7554/eLife.57683.sa1
Author response https://doi.org/10.7554/eLife.57683.sa2

## Additional files

### Supplementary files

- Source data 1. Raw data for western blots.

- Transparent reporting form

## Data availability

ChIP-Seq data has been deposited in GEO under the accession number GSE125960. RNA-Seq data been deposited in GEO under the accession number GSE142975.

The following datasets were generated:

| Author(s) | Year | Dataset title | Dataset URL | Database and Identifier |
|---|---|---|---|---|
| Reshma T, Pal A, Leung JY, Ang GC, Rao VK, Pignata L, Lim HJ, Hebrard M, Chang KT, Lee VK, Guccione E | 2020 | EHMT2 epigenetically suppresses Wnt signaling and is a potential target in embryonal rhabdomyosarcoma | https://www.ncbi.nlm. nih.gov/geo/query/acc. cgi?acc=GSE125960 | NCBI Gene Expression Omnibus, GSE125960 |
| Reshma T, Pal A, Leung JY, Ang GC, Rao VK, Pignata L, Lim HJ, Hebrard M, Chang KT, Lee VK, Guccione E | 2020 | EHMT2 epigenetically suppresses Wnt signaling and is a potential target in embryonal rhabdomyosarcoma | http://www.ncbi.nlm.nih. gov/geo/query/acc.cgi? acc=GSE142975 | NCBI Gene Expression Omnibus, GSE142975 |

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
