## [Decision Letter]

Thank you for sending your article entitled "Deregulation of the histone H3K9 dimethylation landscape suppresses Wnt signaling in embryonal rhabdomyosarcoma" for peer review at *eLife*. Your article has been evaluated by three peer reviewers, and the evaluation was overseen by Maureen Murphy as the Senior and Reviewing Editor.

Summary:

Pal et al. report on G9a inhibition as a potential therapeutic vulnerability in embryonal rhabdomyosarcoma through an indirect suppression of DKK1 via CDK1 mediated phosphorylation of SP1, releasing canonical Wnt signaling and myogenic differentiation. In patient derived cell lines and mouse xenografts, knock down of G9a, through chemical inhibition or siRNA, reduces tumor cell viability. Molecular studies confirm an association of SP1 and P300 on the DKK1(a Wnt signaling antagonist) promoter, and suggest that G9a inhibition is associated with CDK1 activation and phosphorylation of SP1. Phospho-SP1 demonstrates reduced DNA binding affinity and thus the authors suggest that this drives transcriptional down-regulation of DKK1.

The manuscript describes a comprehensive analysis, presented in step-wise and logical fashion. However the main conclusions presented are not as well supported as they could be, and in some instances the results are difficult to follow. Of critical concern is the authors repeated mention, including in the title, that the methyltransferase activity of G9a is directly implicated in the down-regulation of CDK1 – however data in support of this claim are not provided (e.g. H3K9me2 occupancy at CDK1). Additional details are needed to replicate the methods and the experiments appear to be inconsistently performed among the cell lines studied, further weakening the conclusions presented.

Major concerns:

1) Experiments are applied inconsistently among the 3 ERMS cell lines studied across different stages of the analysis. For example, the initial drug screen, where G9a is identified as a target, is performed in JR1 and RD. Colony formation is done in JR1, only whereas shG9a experiments are performed in RD. Finally, DKK1 knockdown and G9a ChIP-seq is performed in RD18. This, in combination with sparse information in parts of the methods, hinders the ability to evaluate findings and thus conclusions drawn.

2) The major finding that G9a inhibition could be an effective treatment option for ERMS, needs to be strengthened. In the drug screen control cell lines are not used, whereas they are used in other aspects of study, for example the HSMM cell is used as normal comparator in the qPCR experiments. Control lines are required to demonstrate specificity of G9a inhibition at concentrations used. This is essential to conclude that G9a activation is of particular relevance to ERMS.

3) Can the authors address: Is loss of Dkk1 expression a direct effect of loss of G9a function at its promoter? Does G9a directly occupy the *Dkk1* promoter?

4) The mechanism for the effect of G9a silencing/inhibition with regard to Sp1 binding to p300 is the weakest part of the manuscript. It appears to be shown in one cell line, using one technique (PLA), for one gene (DKK1). As this is the truly important piece of data (particularly given the extremely modest effect of G9a inhibitors on ERMS xenografts), this mechanistic aspect would need to be significantly improved in a revised version. Specifically: can the authors show this effect by IP-western, with data that include steady state levels of Sp1 and p300? Can they address: does this impact on other Sp1-regulated genes? (If so, why didn't Sp1 come out in the Go analysis of the RNA Seq data?). Also this needs to be repeated in another ERMS cell line. With regard to their PLA data, the authors need the required controls for their PLA, such as the use of only one antibody. Finally, the authors should perform PLA on their “normal control” cells used in Figure 1, to indicate whether this effect is tumor cell specific or not.

---

## [Author Response]

The manuscript describes a comprehensive analysis, presented in step-wise and logical fashion. However the main conclusions presented are not as well supported as they could be, and in some instances the results are difficult to follow. Of critical concern is the authors repeated mention, including in the title, that the methyltransferase activity of G9a is directly implicated in the down-regulation of CDK1 – however data in support of this claim are not provided (e.g. H3K9me2 occupancy at CDK1). Additional details are needed to replicate the methods and the experiments appear to be inconsistently performed among the cell lines studied, further weakening the conclusions presented.Major concerns:1) Experiments are applied inconsistently among the 3 ERMS cell lines studied across different stages of the analysis. For example, the initial drug screen, where G9a is identified as a target, is performed in JR1 and RD. Colony formation is done in JR1, only whereas shG9a experiments are performed in RD. Finally, DKK1 knockdown and G9a ChIP-seq is performed in RD18. This, in combination with sparse information in parts of the methods, hinders the ability to evaluate findings and thus conclusions drawn.

We have now performed CFA with UNC0642 in RD with similar results. The data is shown in Figure 1G.

We created stable G9a knockdown cell line in RD with the aim to use these cells for our in vivo experiments. Most in vivo studies in ERMS have primarily used this cell line. However, we would like to highlight that we have validated the phenotypic effects of G9a knockdown through siRNA based approach in all three ERMS cell lines (RD, RD18 and JR1); with use of UNC0642 in all cell lines as well as using stable lentiviral mediated knockdown in one cell line demonstrating robustness of the data.

Since we saw nearly identical results in all cell lines, we used RD18 for ChIP-seq analysis. As such, DKK1 knockdown experiments were also done in RD18 cells. We will be able to validate the findings in another ERMS cell line to prove robustness. In response to the reviewers, we will move the CDK1 data to the supplementary section and do ChIP-PCR for CDK1 in another ERMS cell line. In the revised version we will also provide detailed methodology for all experiments.

We have done phenotypic analysis of DKK1 knockdown in RD cell line with similar results (Figure 5—figure supplement 1A, B). We have also provided detailed methodology for all experiments. We agree with the reviewer’s comments regarding the CDK1 data and have removed it from the revised manuscript.

2) The major finding that G9a inhibition could be an effective treatment option for ERMS, needs to be strengthened. In the drug screen control cell lines are not used, whereas they are used in other aspects of study, for example the HSMM cell is used as normal comparator in the qPCR experiments. Control lines are required to demonstrate specificity of G9a inhibition at concentrations used. This is essential to conclude that G9a activation is of particular relevance to ERMS.

We tested the efficacy of UNC0642 in normal skeletal myoblasts (HSMM) and ERMS RD cells. It is clear that the G9a inhibitor shows a significantly greater effect on ERMS cell viability compared to normal myoblasts (Figure 1F). As requested by the reviewers, we also provide the IC50 for UNC0642 in HSMM and RD cells in Figure 1E.

3) Can the authors address: Is loss of Dkk1 expression a direct effect of loss of G9a function at its promoter? Does G9a directly occupy the Dkk1 promoter?

We now show G9a occupancy on the *DKK1* promoter and its validation by ChIP-PCR (Figure 4B-C) indicating that DKK1 is a direct G9a target gene. To show specificity of G9a occupancy at the *DKK1* promoter, we have also done ChIP-PCR at regions before and after the G9a peak and did not find any significant G9a enrichment (Figure 4D and E).

4) The mechanism for the effect of G9a silencing/inhibition with regard to Sp1 binding to p300 is the weakest part of the manuscript. It appears to be shown in one cell line, using one technique (PLA), for one gene (DKK1). As this is the truly important piece of data (particularly given the extremely modest effect of G9a inhibitors on ERMS xenografts), this mechanistic aspect would need to be significantly improved in a revised version. Specifically: can the authors show this effect by IP-western, with data that include steady state levels of Sp1 and p300? Can they address: does this impact on other Sp1-regulated genes? (If so, why didn't Sp1 come out in the Go analysis of the RNA Seq data?). Also this needs to be repeated in another ERMS cell line. With regard to their PLA data, the authors need the required controls for their PLA, such as the use of only one antibody. Finally, the authors should perform PLA on their “normal control” cells used in Figure 1, to indicate whether this effect is tumor cell specific or not.

We show by IP-western that Sp1-p300 association is affected in presence of UNC0642 (Figure 4K). Steady state levels of Sp1 and p300 are also shown in the same figure.

We show PLA for Sp1-p300 interaction in RD and HSMM as well in Figure 4—figure supplement 1D-E along with controls for single antibody (Figure 4—figure supplement 1H).

RNA-seq analysis of differentially expressed genes upon G9a knockdown revealed some canonical cancer related SP1 target genes like IGBP2, IGBP3, HB-EGF, *FGFR1*, CCND1, VEGFA and VEGFC. Further Homer analysis of the ChIP seq data for DNA motif enrichment at predicted G9a binding sites revealed KLF7 as one of the top DNA motifs associated with G9a predicted binding sites. SP1 is a member of the Krüppel-like factors (KLFs) all of which share a highly conserved DNA binding domain with high sequence similarity. Thus, SP1 and other KLF family members might play an important role in regulating G9a target genes in ERMS. Since SP1 mRNA levels are not changed (based on G9a RNA-Seq analysis) we believe that reduced SP1/p300 association in presence of G9a inhibitors may lead to reduced SP1 activity and consequently expression of its downstream targets such as DKK1 as well as those listed above. This is in line with reduced SP1 and p300 occupancy at the DKK1 promoter. We have included this in the Discussion and included Homer analysis showing DNA motif enrichment at predicted G9a binding sites (Figure 4—figure supplement 1A).